# Anticancer Activity and Molecular Targets of *Piper cernuum* Substances in Oral Squamous Cell Carcinoma Models

**DOI:** 10.3390/biomedicines11071914

**Published:** 2023-07-06

**Authors:** Thaíssa Queiróz Machado, Maria Emanuelle Damazio Lima, Rafael Carriello da Silva, Arthur Ladeira Macedo, Lucas Nicolau de Queiroz, Bianca Roberta Peres Angrisani, Anna Carolina Carvalho da Fonseca, Priscilla Rodrigues Câmara, Vitor Von-Held Rabelo, Carlos Alexandre Carollo, Davyson de Lima Moreira, Elan Cardozo Paes de Almeida, Thatyana Rocha Alves Vasconcelos, Paula Alvarez Abreu, Alessandra Leda Valverde, Bruno Kaufmann Robbs

**Affiliations:** 1Postgraduate Program in Applied Science for Health Products, Faculty of Pharmacy, Fluminense Federal University, Niteroi 24241-000, RJ, Brazil; thaissaqm@id.uff.br (T.Q.M.); lucasnicolaunf@gmail.com (L.N.d.Q.); 2Department of Organic Chemistry, Chemistry Institute, Fluminense Federal University, Niteroi 24020-141, RJ, Brazil; mariaemanuelle.damazio@gmail.com (M.E.D.L.); biancaangrisani@id.uff.br (B.R.P.A.); thatyanavasconcelos@id.uff.br (T.R.A.V.); 3Postgraduate Program in Dentistry, Health Institute of Nova Friburgo, Fluminense Federal University, Nova Friburgo 28625-650, RJ, Brazil; faelcarriello@gmail.com (R.C.d.S.); fonseca.anna@gmail.com (A.C.C.d.F.); 4Pharmaceutical Sciences, Food and Nutrition Faculty, Mato Grosso do Sul Federal University, Campo Grande 79070-900, MS, Brazil; arthur.ladeira@ufms.br (A.L.M.); carlos.carollo@ufms.br (C.A.C.); 5Basic Science Department, Health Institute of Nova Friburgo, Fluminense Federal University, Nova Friburgo 28625-650, RJ, Brazil; priscilla_rodrigues@id.uff.br (P.R.C.); elancardozo@id.uff.br (E.C.P.d.A.); 6Biodiversity and Sustainability Institute, Macaé Campus, Federal University of Rio de Janeiro, Macae 21941-901, RJ, Brazil; vitorwrabelo@gmail.com (V.V.-H.R.); abreu_pa@yahoo.com.br (P.A.A.); 7Research Directorate, Laboratory of Natural Products and Biochemistry, Rio de Janeiro Botanical Garden Research Institute, Rio de Janeiro 22460-030, RJ, Brazil; davysonmoreira@hotmail.com

**Keywords:** oral cancer, Piperaceae, GNPS, antitumor, apoptosis, isocorydine, boldine, asimilobine, coclaurine, *N*-methylcoclaurine

## Abstract

Oral squamous cell carcinoma (OSCC) is a worldwide public health problem, with high morbidity and mortality rates. The development of new drugs to treat OSCC is paramount. *Piper* plant species have shown many biological activities. In the present study, we show that dichloromethane partition of *Piper cernuum* (PCLd) is nontoxic in chronic treatment in mice, reduces the amount of atypia in tongues of chemically induced OSCC, and significantly increases animal survival. To identify the main active compounds, chromatographic purification of PCLd was performed, where fractions 09.07 and 14.05 were the most active and selective. These fractions promoted cell death by apoptosis characterized by phosphatidyl serine exposition, DNA fragmentation, and activation of effector caspase-3/7 and were nonhemolytic. LC–DAD–MS/MS analysis did not propose matching spectra for the 09.07 fraction, suggesting compounds not yet known. However, aporphine alkaloids were annotated in fraction 14.05, which are being described for the first time in *P. cernuum* and corroborate the observed cytotoxic activity. Putative molecular targets were determined for these alkaloids, in silico, where the androgen receptor (AR), CHK1, CK2, DYRK1A, EHMT2, LXRβ, and VEGFR2 were the most relevant. The results obtained from *P. cernuum* fractions point to promising compounds as new preclinical anticancer candidates.

## 1. Introduction

Oral squamous cell carcinoma (OSCC) is among the most prevalent malignant neoplasms worldwide and usually has a poor prognosis. This disease has a high incidence and even with the advances already achieved in diagnosis and treatment, morbidity and mortality rates are still high, especially after the occurrence of metastasis. After primary treatment, relapses and/or metastases are found in more than half of the patients (80% of cases in the first 2 years), and the 5-year survival rate is still less than 50%, resulting in a serious issue for public health [1,2,3,4,5,6]. The demographic distribution, incidence, and patient survival have changed very little over the past 30 years. These factors strengthen the need to search for new treatments and improvements in diagnosis [7]. The identification of new natural compounds that can halt tumor growth with a minimal negative impact on human health is crucial from both clinical and scientific perspectives [8].

*Piper* L. species have been extensively studied due to their biological activities in many in vitro and in vivo experiments. These species have demonstrated outstanding potential as antiproliferative agents [9]. *Piper cernuum* is a shrub native to Brazil, reaching a height of approximately 6 m and featuring large leaves measuring up to 40 cm. It is found in both primary and secondary forests and is commonly known as “pimenta de macaco” (monkey pepper). The leaves of this plant are used to prepare infusions and decoctions for the treatment of kidney and liver diseases, ulcers, colds, fevers, bronchitis, and urinary tract infections [10]. In recent efforts of our research group in this field, we have examined five species of the genus *Piper* against OSCC cell lines. The dichloromethane partition of *Piper cernuum* Vell. leaves (PCLd) revealed encouraging in vitro findings, showing high cytotoxicity and selectivity against OSCC and low toxicity in mice [11].

Our goal is to study the effects of a chromatographic fraction from PCLd on oral squamous cell carcinoma and fibroblasts, specifically looking at its cytotoxicity and selectivity. To evaluate its potential as a treatment, we conduct tests on in vivo OSCC models, analyzing chronic toxicity and survival rates. Additionally, we utilize the GNPS platform to annotate primary compounds and perform ^1^H NMR analysis. Finally, we explore potential molecular targets through in silico molecular docking.

## 2. Materials and Methods

### 2.1. Plant Material

Leaves of *Piper cernuum* Vell. (PC—leaves) were harvested in January 2018 at Parque Nacional da Serra dos Órgãos, Guapimirim/RJ. A sample of the plant material was sent to Dr. Elsie Franklin Guimarães, a specialist in Piperaceae, who identified the plant. This study was authorized by the Chico Mendes Institute of Biology (ICMBio) through the Biodiversity Authorization and Information System (SISBIO) in document number 45566-1. Access was authorized by the Genetic Heritage Management Council (CGEN) in process 010771/2014-0.

### 2.2. Plant and Crude Extract Preparation

Extract preparation was carried out in accordance with Macedo et al. (2019) and [11], as well as PCLd obtention. Briefly, the dried crushed leaves were extracted by maceration until exhaustion with methanol that was evaporated in a rotatory evaporator under reduced pressure. The extract was reconstituted in methanol:water (1:1) and extracted with hexane, dichloromethane, and ethyl acetate, successively.

### 2.3. PCLd Fractionation

The dichloromethane partition (PCLd) was separated into seventeen fractions by vacuum liquid chromatography (VLC) [12] with silica gel with 5–25 µm particle size (Sigma-Aldrich^®^, Burlington, MA, USA) and using a funnel with a sintered plate measuring 9.5 × 9.0 cm. The sintered funnel was filled under vacuum and eluted in a polarity gradient, with hexane, ethyl acetate, methanol, and water. In the selected fractions (09 and 14), activated charcoal was added to the mass of the sample (proportion 1:1) and the amount of methanol was used in a proportion of 100:1 to the mass of activated charcoal [13] to remove chlorophyll. Then, VLC was performed to fractionate 09 and 14, using, in both procedures, a funnel with a sintered plate measuring 2.5 × 5.0 cm and dichloromethane and methanol as eluents. Seventeen fractions were obtained from fraction 09, and twenty fractions were obtained from fraction 14.

### 2.4. LC–DAD–MS/MS Analyses

For the analyses of fractions 09.07 and 14.05, an ultra-fast liquid chromatography (UFLC) Shimadzu Prominence chromatographic system coupled to a diode array detector (DAD) and an electron-spray ionization (ESI) mass spectrometer (MicrOTOF-Q III, Bruker Daltonics, Billerica, MA, USA) was used. The chromatographic separations were made in a Kinetex C18 chromatographic column (150 mm × 2.1 mm i.d., 2.6 μm, 100 Å, Phenomenex), with a constant flow rate at 0.3 mL/min, at 50 °C. The DAD monitored wavelength between 240 and 800 nm and the tandem mass spectrometer operated in negative and positive ion modes (*m*/*z* 120–1200). An experimental blank was analyzed in the same conditions. All the LC and MS parameters were the same as described by Macedo et al. (2021) [14]. The molecular formulae were determined based on the accurate mass (±5 ppm) and mSigma below 30.

### 2.5. Analysis of Mass Spectra and Annotation Using GNPS

Data acquired by LC–DAD–MS/MS were converted to .mzML format with MSConvert software (Version 3.0) [15], then uploaded through the FTP protocol with the WinSCP^®^ software Version 5.19.2. Subsequently, the dereplication of these data was performed online in the Global Natural Products Social (GNPS) using the Molecular Network tool [16]. The data were entered into the platform in different groups to allow a graphic representation, with the nodes referring to fraction 14.05 in group 1 with pink coloring, and the nodes of fraction 09.07 in group 2 with blue coloring. The pink and blue nodes are shown in different proportions that represent the abundance of the signs in each fraction. The analysis parameters were tolerance of precursor ion mass at 0.03 Da; EMn ion fragment tolerance of 0.03 Da; matches maintained between network or library spectra with cosine score above 0.65; and at least 3 fragments of concordant mass. The data created in the GNPS were then transferred to Cytoscope software (Version 3.8.0) for visualization [17]. Some compounds were also annotated by dereplication using MassBank (http://massbank.eu/MassBank (accessed on 24 September 2021)). Signals that were present in the experimental blank were removed during the analysis of the samples.

### 2.6. NMR Analysis

NMR spectra were recorded on a Varian VNMRS 500 spectrometer operating at 500.00 MHz (^1^H) in deuterated DMSO. Chemical shifts are reported in ppm (δ) relative to tetramethylsilane (TMS).

### 2.7. Mouse Husbandry

Female C57Bl/6 mice (n = 64) with a maximum age of 12 weeks were obtained from the vivarium of UFF-Niteroi-RJ. All animals were housed in standard polypropylene cages at a controlled temperature (25 ± 2 °C), with light conditions (12 h of light and dark cycle) and relative humidity (65 ± 5%). The animals received diet and water ad libitum. All experimental protocols were approved by the Animal Ethics Committee under the number CEUA/UFF 982. All experiments were performed under Brazilian guidelines and regulations.

### 2.8. Chronic Toxicity, and Therapeutical Properties of PCLd and Survival Analysis

Female C57Bl/6 mice (n = 64) with a maximum age of 12 weeks were used. The induction of carcinogenesis in the mice tongues was performed using 4-Nitroquinoline 1-oxide (4NQO) (Sigma, St. Louis, MO, USA) carcinogen dissolved in ethylene glycol (Sigma) and placed in the drinking water of the animals at the final concentration of 100 μg/mL as previously described ([18]). The average water intake with or without 4NQO, per cage, was recorded throughout the experimental period and there was no difference between the groups.

For chronic toxicity and therapeutical analysis, the animals were divided into 4 groups: PBS (control; n = 6), PCLd (n = 6), 4NQO (n = 6), and 4NQO + PCLd (n = 6). In the first 8 weeks, 4NQO (100 µg/mL) was administered in the drinking water for the groups 4NQO or 4NQO + PCLd. Afterward, an eight-week interval was given for tumors to develop, which were present in all animals treated with 4NQO, as determined by visual inspection. From the 16th week, therefore, PCLd was administered in indicated groups by intraperitoneal injection at 30 mg/kg twice a week for eight weeks (totaling 24 weeks of experiment and a total of 480 mg/kg of PCLd per animal) for the groups PCLd and 4NQO + PCLd. In the 24th week, the animals were euthanized through cervical dislocation, and organs/tissues were collected. The main organs and tongues were identified and photographed for macroscopic analysis.

### 2.9. Macroscopic and Histopathological Analysis of the Tongue

At the end of the 24th week, tongues from animals were collected for macroscopic and histopathological analysis. These analyses were made independently by three oral pathologists: Dr. Rebeca S. Azevedo, Dr. Priscilla R. Câmara, and Ms. Rafael C. da Silva, who, after the blind analyses, consolidated their findings. In the macroscopic analysis, based on the characteristics of the lingual surface, the following aspects were taken into account: (1) Normal: smooth, regular surface, and pinkish color (normochromic); (2) increase of volume: elevated, solid, firm, pedunculated or sessile and well-circumscribed lesions (nodules) and finger-like or cauliflower-like projections, with sessile base (papillomatous lesions); (3) color change: white lesions (flat or thick, spot or plaque, which may be translucent, fissured, or wrinkled—leukoplakia) and red lesions (spot or plaque, erythematous with a velvety texture—erythroplakia); (4) ulcer: discontinuity of the lingual surface); (5) alteration of the midline; (6) irregular borders.

For the histopathological analysis, the tongues were cut in the longitudinal direction, and the halves were included according to the orientation of the desired section. The fragments underwent processing similar to that of the chronic toxicity section. Then, 3 μm thick cuts were made and for every 10 sequential cuts, 1 cut was used for staining with hematoxylin–eosin (HE) until the final wear of the block. About 500 slides were generated. The slides were observed on the Nikon Eclipse E-200 (Nikon, Tokyo, Japan) microscope and the changes were quantified taking into account the criteria proposed by WHO in its most recent classification. In this study, the following classification was used: (1) hyperplasia (2) mild dysplasia; (3) moderate dysplasia; (4) severe dysplasia/carcinoma in situ; (5) carcinoma.

### 2.10. Chronic Toxicity of PCLd

For the study of chronic toxicity, the following signs were used as an indication of morbidity: tremors, convulsion, salivation, diarrhea, lethargy, coma, signs of pain, increased back arching, and mobility defect, as described in Zorzanelli, 2022 [19]. Fragments of the lungs, heart, spleen, liver, and kidneys of animals were collected and fixed in a 10% buffered formalin solution, dehydrated in decreasing dilutions of alcohol, cleared, and embedded in paraffin. The blocks were cut into 5 μm sections and the sections were stained with routine HE staining. The slides were observed on the Nikon Eclipse E-200 microscope and the morphological changes of each organ were analyzed.

### 2.11. Survival Analysis

The experimental design used was identical as beforementioned for chronic toxicity but we extended the time to analyze animal survival rate. The animals were divided into four groups: PBS (control; n = 10), PCLd (n = 10), 4NQO (n = 10), and 4NQO + PCLd (n = 10). After the end of treatment with PCLd in the 24th week, survival of mice was observed until the 52nd week (one year). As a mortality criterion, the animals were considered dead when they effectively died or reached a body weight of less than half the average weight of the control group. For statistical survival analysis, a Kaplan–Meier survival curve was used with curve comparison using Logrank (Mantel–Cox test) on GraphPad 8.0 (Intuitive Software for Science, San Diego, CA, USA).

### 2.12. Cell Viability Assay (Cytotoxicity)

The semipurified fractions derived from the dichloromethane partition were tested for their cytotoxicity in tumor and normal cells. Their biological activity was compared regarding selectivity and used to guide further downstream fraction selection. SCC9 cells, derived from human tongue OSCC, were obtained from (ATCC^®^ CRL-1629 and CRL-1624) and maintained in 1:1 DMEM/F12 medium (Dulbecco’s modified Eagle’s medium and Ham’s F12 medium; Gibco (Thermo Fisher, Waltham, MA, USA) supplemented with 10% (*v*/*v*) FBS (Fetal Bovine Serum; Invitrogen, Thermo Fisher, Waltham, MA, USA) and 400 ng/mL hydrocortisone (Sigma-Aldrich Co., St. Louis, MO, USA). Human primary normal gingival fibroblasts were also obtained from ATCC and maintained in DMEM supplemented with 10% (*v*/*v*) FBS. The cells were cultured in a humidified environment containing 5% CO_2_ at 37 °C. The viability of OSCC cells and primary fibroblasts were tested by the MTT assay as described in Fonseca, 2021 [20]. Cells were seeded in triplicate in 96-well plates (5 × 10^3^ cells/well) until confluency. Then, the medium was removed, fresh medium was added, and the cells were returned to the incubator in the presence of the indicated fraction. The chromatographic fractions (3, 5, 6, 9, 10, 12, and 14) obtained from the dichloromethane partition were tested for cell viability on SCC9 cells and fibroblasts. Fractions 9 and 14 were the most selective and were further fractioned, giving rise to fractions 09.01, 09.03, 09.05, 09.07, 09.09, 14.03, 14.05, 14.07, 14.09, and 14.10, which were also tested on SCC9 cells and fibroblasts. Chromatographic fractions were solubilized in DMSO (Sigma-Aldrich, St. Louis, MO, USA) to a final concentration of 25 mg/mL. Other concentrations were prepared from this dilution to perform the cell viability assay. Carboplatin (Fauldcarbo^®^; Libbs Farmacêutica, São Paulo, SP, Brazil) was used as an antitumor control. DMSO was used as a negative control, representing 100% cell viability. After 48 h of treatment, cells were incubated for 3.5 h with 5 mg/mL MTT reagent (3,4,5-dimethiazol-2,5-diphenyltetrazolium bromide) (Sigma-Aldrich Co., St. Louis, MO, USA) diluted in culture medium. Subsequently, formazan crystals were dissolved by MTT solvent (50% Methanol and 50% DMSO), and the absorbance was read at 560 nm using the EPOCH microplate spectrophotometer (BioTekInstruments, Winooski, VT, USA). Each fraction was tested in at least seven different concentrations, varying from 15.6 μg/mL to 1000 μg/mL.

### 2.13. Statistical Analysis, Calculation of IC_50_, and Selectivity Index (SI)

When indicated, data were tested by one-way ANOVA with Tukey’s posttest with multiple comparations (comparing each column with the mean of every other column). The analyses were conducted using GraphPad Prism Software Version 5.0, and a *p* < 0.05 was considered statistically significant.

IC_50_ values for the MTT assay were obtained by nonlinear regression using GraphPad 8.0 (Intuitive Software for Science, San Diego, CA) from at least three independent experiments. Data are presented as means ± standard deviation (SD). A log dose response (inhibitor) vs. response curve using the least square method was used to determine the IC_50_ and SD of the data. The selectivity index was calculated as SI = IC_50_ of the compound in fibroblasts/IC_50_ of the same compound in OSCC cells.

### 2.14. ROS Production

The hydrogen peroxide (H_2_O_2_) luminescence detection assay was performed using the ROS-GloTM H_2_O_2_ Assay Kit (Promega, Madison, WI, USA). To perform the experiment, in a 96-well plate, 1 × 105 SCC9 cells were seeded in DMEM F12 with 10% FBS medium per well. The cells were incubated for 24 h and, after this period, the medium was removed and the cells were treated with 2 × IC50 from the 09.07 and 14.05 fractions or with the DMSO control at 6, 24, and 48 h. Cell-free wells were used as a control. Menadione (M9529-Sigma-Aldrich, St. Louis, MO, USA) was used as a positive control. After the indicated time, the H_2_O_2_ substrate was used for all treatments. After that, the detection solution was added and the plates were read in the luminometer (Turner Designs, TD 20/20, San Jose, CA, USA).

### 2.15. Hemolysis Assay

Human blood was used in the hemolysis assay, as approved by the Research Ethics Committee of the Fluminense Federal University—Nova Friburgo-RJ (CAAE: 43134721.4.0000.5626). Erythrocytes were collected by centrifugation at 1500 rpm for 15 min, washed with PBS (phosphate buffer saline) with 10 mM glucose, and counted in an automatic cell counter (Thermo Fisher, Waltham, MA, USA). Erythrocytes were seeded in 96-well plates at a concentration of 4 × 10^8^/well in duplicate. *Piper cernuum* fractions were at a concentration of 500 μg/mL in PBS with glucose (100 μL final volume). PBS was used as a negative control and 0.1% Triton ×100 was used as a positive control. Data reading was performed with EPOCH (BioTek Instruments, Winooski, VT, USA) at 540 nm absorbance, and statistical data were generated with the GraphPad Prism 5.0 program (Intuitive Software for Science, San Diego, CA, USA).

### 2.16. Cell Cycle and Analysis of SubG1

To evaluate the action of fractions 09.07 and 14.05 on the cell cycle, cells of the SCC9 lineage were plated in a 6-well plate (5 × 10^5^ cells/well) as described in [21]. After 48 h of treatment, cells were trypsinized and stained with propidium iodide (75 µM; Sigma, St. Louis, MO, USA) in the presence of NP-40 (Sigma). DNA content was analyzed by collecting 1 × 10^4^ events using a FACScalibur flow cytometer. Data were analyzed using CellQuest (BD Biosciences, Franklin Lakes, NJ, USA) and FlowJo (Version 9.9.4; Tree Star Inc., San Carlos, CA, USA) software.

### 2.17. Analysis of Exposure to Phosphatidylserine (Apoptosis)

Cells of the SCC9 lineage were plated in 6-well plates (5 × 10^5^ cells/well), trypsinized 48 h after treatment with fractions 09.07 and 14.05, labeled with the Annexin V-FITC Apoptosis Detection Kit according to the manufacturer’s protocol (BMS500FI, Invitrogen, Thermo Fisher, Waltham, MA, USA), and analyzed by flow cytometry, as described in [22].

### 2.18. Active Caspase 3/7 Assay

Cells of the SCC9 lineage were plated in 24-well plates (5 × 10^5^ cells/well), treated with fractions 09.07 and 14.05 (2 × IC_50_), and with caspase marker (CellEvent™ Caspase-3/7 Green ReadyProbes™ Reagent Invitrogen, Thermo Fisher, Waltham, MA, USA). After that, the cells were trypsinized and analyzed by flow cytometry.

### 2.19. Prediction of the Mechanism of Action of the Main Compounds of the 14.05 Fraction

This method is described in more detail in Supplementary Data. We used different molecular modeling strategies based on the five main phytochemicals found in the fraction to predict the mechanism of action of the compounds responsible for the biological activity. Briefly, compounds were separated into two groups according to their chemical similarity. A similarity search was conducted within PDB and ChEMBL databases to select potential targets for these compounds. Second, a pharmacophore search was performed using the Pharm Mapper webserver. A normalized fit score of 0.9 was used as the cutoff and compounds must have fulfilled all the pharmacophore criteria. Finally, for both approaches, proteins were chosen according to experimental evidence in the literature that correlates them with anticancer effects, particularly in OSCC. After target selection, inverse docking studies were carried out. Three-dimensional structures of the compounds were constructed using RDKit and further submitted to geometry optimization using OpenBabel 3.1 (http://openbabel.org/ (accessed on 24 September 2021)). Proteins were prepared in Autodock Tools 1.5.7 or Hermes 2022.3. Different docking protocols and programs were used and they are summarized in Appendix A. All docking protocols were validated by redocking studies, and RMSD values of the top-scoring poses were lower than 2 Å (Appendix A). The top-scoring pose of each ligand with each protein was selected for scoring normalization and interaction analysis using Pymol 2.5 (The PyMOL Molecular Graphics System, Version 2.5, Schrödinger, LLC) and Discovery Studio Visualizer 2021 (Dassault Systèmes BIOVIA, San Diego, CA, USA, 2021). Docking scores obtained from Autodock 4.2.6 were negated; then, the values were normalized by using the combined Z-score method [23] for comparison among the different targets.

## 3. Results and Discussion

### 3.1. Piper cernuum Partition Is Nontoxic, Reduces Cancer Foci Numbers and Size, and Increases the Survival of Mice with Chemically Induced OSCC

In previous results from our group, we determined that dichloromethane fraction from *Piper cernuum* leaves (PCLd) was highly cytotoxic and selective towards oral squamous carcinoma cell lines and tolerated in mice (in vivo) in the acute toxicity test [11].

To deepen our knowledge about the possible use of these plants in cancer treatment, a therapeutic application of PCLd was performed in oral cancer based on an animal chemically induced model using 4NQO. Previous reports by several groups have shown that 4NQO induces oral SCC being a preclinical model for the study of OSCC [24,25,26]. 4NQO can produce OSCC in mice [27,28], inducing the formation of DNA adducts and resulting in the replacement of adenosine by guanosine [29] and producing a progression model of temporal carcinogenesis that demonstrates multiple hyperplastic, dysplastic, and neoplastic lesions after prolonged treatment [28]. In our work, OSCC in the tongue was chemically induced in C57Bl/6 animals 4NQO. The experimental design for the bioassay is illustrated in Figure 1A.

The main macroscopic alterations at the tongue were analyzed and counted (Figure 1B,C), chronic toxicity of 4NQO and PCLd treatment were assessed (Table 1; Appendix A), and histopathology analyses of the main hyperplastic, dysplastic, and carcinomas alteration at the tongue were evaluated (Figure 2).

After the therapeutic application of PCLd in OSCC of chemically induced animals, a macroscopic analysis of the tongue of these mice was performed (Figure 1A). Representative images of the tongue of each group were taken and are shown in Figure 1B. All animal tongues were analyzed macroscopically, based on the characteristics of the alteration described in methods that were quantified (Figure 1C). As expected, the groups that received only PBS and PCLd showed no observable macroscopical clinical changes. However, the 4NQO and 4NQO + PCLd groups presented lesions or alterations classified as nodules, papillomatous lesions area, erythroplakia, leukoplakia, ulcers, and more global alterations as midline deviation and irregular borders (Material and Methods). These alterations occurred in greater size (Figure 1B) and numbers (Figure 1C) in the 4NQO group compared to the 4NQO + PCLd group. The most pronounced effect of PCLd treatment was in reducing the number of papillomatous lesions and color changes (erythroplakia or leukoplakia), where the latter can be considered potentially malignant lesions (Figure 1C). Furthermore, 4NQO induced irregular borders (100% of the animals) and midline shifts (50% of the mice), where PCLd treatment apparently normalized these alterations in all animals (Figure 1B).

Chronic toxicity studies aim to characterize the toxicological profile of a substance through repeated administration in animals. Thus, it is possible to obtain information on toxic effects, identification of target organs, and effects on the animal’s physiology [30]. Analyzing the chronic treatment with PCLd or 4NQO in mice after 24 weeks (Figure 1A), it was observed that the animals did not show an increase in mortality or clinical signs of toxicity in the skin, eyes, mucous membranes, gait, secretions, excretions, autonomic activity, posture management, clonic or tonic movements, or stereotypes and bizarre behavior over the course of treatment (Table 1). They also showed no reduction in weight and food intake. Examination of histological sections of organs (lungs, heart, spleen, liver, and kidney) did not show alterations considered specific for PCLd intoxication (Table 1). The architecture of these organs was also well preserved. 4NQO treatment induced pulmonary arterial/venous and portal hyperemia, and the appearance of lymphocyte focused on the liver, where PCLd treatment reduced perivascular and periportal lymphocyte focus (Table 1 and Appendix A). The results show no apparent limiting toxic effects of PCLd in mice at the tested concentration, making this fraction a good candidate for other in vivo anticancer tests.

We conducted a histopathological analysis of mouse tongue tissues to investigate the effect of PCLd treatment on the development of OSCC induced by 4NQO (Figure 1A and Figure 2). The histological analysis provides valuable information on the presence of benign or malignant lesions, clinical behavior, and prognostic information about the lesions [31,32,33]. The histological slides with normal or atypical findings are shown in Figure 2A as representative images, which were characterized according to the WHO guidance. Microscopic analysis was consistent with macroscopic findings and showed a statistical difference with *p* < 0.05 for moderate dysplasia and squamous cell carcinoma between the 4NQO and 4NQO + PCLd groups (Figure 2B). Therefore, mice in the 4NQO group had a higher average number of lesions with pronounced epithelial alterations than those in the 4NQO + PCLd group, indicating that PCLd may reduce the formation of more severe oral lesions.

We proceeded with a survival rate assay to understand the life expectancy of mice treated with 4NQO and the influence of PCLd. A Kaplan–Meier survival curve was generated, and the experiment is described in Figure 3A. Mice in control, PCLd, 4NQO, and 4NQO + PCLd groups were monitored for 52 weeks (1 year). The 4NQO group exhibited a median survival of 40.5 weeks, while animals in the 4NQO + PCLd group had a median survival of more than 52 weeks (Figure 3B,C). This result indicates a survival increase of over 20% in the PCLd-treated animals and a significant (*p* < 0.0001) improvement in survival compared to those without treatment.

In addition to the survival benefit, PCLd demonstrated a decrease in the number of potentially malignant lesions and restored the tongue midline displacement in mice with OSCC. Histological examination of the animals’ organs showed no signs of toxicity after chronic treatment with PCLd. These findings prompted us to conduct further in vitro studies with PCLd-derived fractions.

### 3.2. Cytotoxicity and Selectivity Determination of PCLd of Fractions

As the in vivo treatment with the PCLd showed a decrease in lesions on the tongue with a lower degree of cellular atypia and severity, a significant increase in overall survival, and an absence of toxicity in mice, we started to investigate the major compounds present in PCLd that might account for these effects. Thus, the PCLd was subjected to fractionation using the VLC technique obtaining 18 fractions. Among them, fractions 3, 5, 6, 9, 12, and 14 were selected to be tested for cell viability in SCC9 cells based on the criteria of polarity difference, a larger amount of sample, and lower complexity according to the chromatographic profile observed by thin-layer chromatography. We used the SCC-9 cell line to analyze the antitumor activity and primary normal human gingival fibroblasts to calculate the selectivity index as previously described by our group [11,20]. As shown in Table 2, fractions 9 (IC_50_ = 40.25 ± 0.06) and 12 (IC_50_ = 45.08 ± 0.04) were the most active compared to the positive control carboplatin (IC_50_ = 322.30 ± 0.04) which is the drug commonly used in the clinic. We and others have already shown that *Piper cernuum* extracts exhibit cytotoxicity in various human and murine tumor cells [11,34].

Selectivity was determined in normal fibroblasts, where fractions 9 (SI = 2.67) and 14 (SI > 7.7) were the most selective, while the carboplatin showed lower selectivity than the tested fractions (SI = 0.99) (Table 2). From these data, fractions 9 (most cytotoxic) and 14 (most selective) were selected to undergo a new chromatographic separation using the same technique (Table 2, bottom). Seventeen fractions of 9 and twenty fractions of 14 were obtained. The same selection criteria of the previous fractions were used to select the fractions to be submitted to the cell viability test in SCC9 cells and normal fibroblasts. The most active and selective fraction among fraction 9 was 09.07 (IC_50_ = 36.87 ± 0.01/SI = 2.03) and among fraction 14 was 14.05 (IC_50_ = 64.2 ± 0.04/SI = 2.53) (Table 2, bottom side). Therefore, fractions 09.07 and 14.05 had high cytotoxicity and selectivity when compared to carboplatin.

### 3.3. Cell Death Pathway Investigation

To analyze the usefulness of these fractions in preclinical tests and to discard the possibility of acting as a surfactant, we performed in vitro hemolysis tests. Fractions 3, 5, 6, 9, 12, 14, 9.07, and 14.05 and the controls did not promote hemolysis in human red blood cells, as well as all other first fractions and controls, being less than 5%, a percentage considered tolerable (Figure 4A). These results make them all suitable for future tests on animals.

Based on the findings that fractions 09.07 and 14.05 exhibited the highest selectivity, our attention now shifts to investigating the underlying mechanism and pathway responsible for cell death. Chemotherapy can trigger various types of cell death, and pinpointing the specific pathway is crucial in the development of novel anticancer treatments [35]. We observed that fractions 09.07 and 14.05 did not induce a significant release of ROS at the times tested, showing lower values than the negative control DMSO (Figure 4B). The positive control, menadione, produced large amounts of ROS, demonstrating that these cells can produce ROS. Further, treatment of SCC9 cells with both fractions (09.07 and 14.05) showed increased single (Annexin V) and double (Annexin V + PI) staining (Figure 4C), induced DNA fragmentation in >18% of the cells (Figure 4D, Sub-G1 DNA-content), and activated effector caspase 3/7 labeling (Figure 4E). Altogether, the results demonstrate that both fractions promote oral cancer cell death by apoptosis. These fractions’ treatment observed no significant change in cell cycle compared to the control (Figure 4F). These data agree with the study that shows increased cleaved caspase-3 in the tumor tissue of human prostate xenografts treated with *Piper betel* leaf extract [36]. Further, ethanol extracts from black pepper (*Piper nigrum* L. cv. Bragantina) showed increased apoptosis of Ehrlich ascites tumor cells [37].

### 3.4. Phytochemical Analysis

The most promising fractions (09.07 and 14.05) were selected for compound annotation by LC-MS/MS with the support of GNPS spectral libraries. The annotation of the compounds in the molecular network was performed based on the exact mass and mass fragmentation pattern.

A total of 18 molecular families were obtained by grouping the spectra for both fractions. The fractions presented diverse chemical profiles, with only nine peaks occurring in both. Furthermore, the clustering of nodes suggests different chemical classes in the fractions, as most of the interactions occurred between compounds of the same samples (Appendix A).

As a result of the chemical analysis, we highlight a molecular family composed of five nodes, all present only in fraction 14.05, the most selective fraction that induces higher caspase-3 activation (Table 2 and Figure 4E): three aporphine alkaloids, and two benzyltetrahydroisoquinoline alkaloids. Two aporphine alkaloids with molecular formulae C_20_H_23_NO_4_ (*m*/*z* 342.168) and C_19_H_21_NO_4_ (*m*/*z* 328.154) were annotated as isocorydine (**1**) and boldine (**2**), respectively (Figure 5). The mass spectrum and the mirror of the substance library are shown in the Appendix A). Their fragmentation spectra presented the base peak with *m*/*z* 265 that represents the opening of the isoquinoline ring and loss of the methylamino group, a characteristic fragmentation of aporphine alkaloids, followed by the formation of a furan-type intermediate and loss of a neutral methyl fragment (CH_3_·) in **1** or the formation of an aromatic epoxide by the loss of MeOH in **2** [38].

The third aporphine alkaloid presented molecular formulae of C_17_H_17_NO_2_ (*m*/*z* 268.131). The library of GNPS suggested the annotation as apomorphine, a tertiary amine. However, the evaluation of the fragmentation spectra showed a loss of 17 *u*, characteristic in aporphine alkaloids with a secondary amine group [38]. In this way, we annotated this alkaloid as the positional isomer asimilobine (**3**) (Figure 5). The benzyltetrahydroisoquinoline alkaloids presented molecular formulae of C_17_H_19_NO_3_ (*m*/*z* 286.144) (**4**) and C_18_H_21_NO_3_ (*m*/*z* 300.159) (**5**). Both mass spectra (Appendix A) showed a base peak at *m*/*z* 209, representing the loss of amino or methylamino groups, respectively, followed by epoxide formation and loss of CO. The additional findings of fragments with *m*/*z* 178 in **4** and *m*/*z* 192 in **5** mass spectra, indicate the loss of a hydroxybenzyl group from [M+H]^+^, a characteristic fragmentation of benzyltetrahydroisoquinoline alkaloids [39]. In this way, these compounds were annotated as coclaurine (**4**) and *N*-methyl coclaurine (**5**). Together, **4** and **5** represented 1.5% of the total chromatogram area.

The ^1^H NMR spectrum also corroborated the presence of these compounds. Analyses of the fraction 14.05 showed simplets at 6.43, 6.74, and 6.79 ppm, which are chemical shifts characteristic of aromatic hydrogens. These data are in accordance with the literature for isocoridine (**1**) at 6.70 ppm, boldine (**2**), and asimilobine (**3**) at 6.64 ppm and to coclaurine (**4**) at 6.65 ppm. For boldine (**2**), an aromatic simplet is also reported at 6.83 ppm to hydrogen H-8 of the basic aporphine alkaloid [40,41,42].

None of these compounds was previously reported in the genus *Piper*. The anticancer activity of all compounds **1**, **2**, **3**, **4**, and **5** have already been reported in in vitro and in vivo experiments, indicating that these compounds may play an important role in the activities described in the present work. Compound **1** demonstrated activity against human hepatocellular carcinoma [43,44,45]). Similarly, boldine (**2**) has broad cytotoxic activity against breast, colon, and neuroblastoma cancer [46,47,48,49,50]. Asimilobine (**3**) was reported as active against prostate and stomach cancer [51]. Compound **4** is active against colon, breast, and liver cancer, and **5** can inhibit the enzyme disulfide isomerase, which supports the progression of several cancers [52,53].

The GNPS platform did not provide spectra match for fraction 09.07. A search using the molecular formulas in the SciFinder database and rationalization of the possible fragmentation mechanisms of the main peaks in this fraction suggested the presence of compounds that have not yet been identified.

### 3.5. Prediction of the Molecular Targets of Annotated Alkaloids in PCLd 14.05 Fraction

To investigate the mechanism of action for the cytotoxic effect against cancerous cell lines of the fractions, we employed an in silico target fishing strategy for the annotated new main phytochemicals compounds of this fraction. Due to their chemical similarities, boldine, isocorydine, and asimilobine were grouped and the first one was selected as the representative compound, while in contrast, coclaurine and *N*-methylcoclaurine were grouped, coclaurine being the representative one. We conducted a similarity search of the representative compounds and searched the representative compounds similarly using the ChEMBL and PDB databases. In addition, a pharmacophore search using a human protein target database was carried out using the PharmMapper webserver. Finally, we selected seven proteins that have been previously associated with cancer, especially OSCC, and might be targeted by these phytochemicals to trigger anticancer activity (Table 3).

Furthermore, molecular docking studies were conducted to evaluate the theoretical affinity of the compounds of the 14.05 fraction with the seven proteins. Coclaurine enantiomers exhibited higher affinity with AR and VEGFR2, even in comparison to the proven target EHMT2 (Figure 6). It is important to highlight that many multitarget therapies have been investigated for several complex diseases, including cancer [63,64]. By contrast, *N*-methylcoclaurine enantiomers had the highest affinity with CK2, CHK1, and EHMT2. Also, these isomers showed positive and high Z-score values with VEGFR2. Despite structural similarities among boldine analogs, we observed different affinity profiles with this panel of proteins, and the highest Z-score values were observed for CK2, CHK1, LXRβ, and EHMT2. These compounds also showed higher Z-scores with DYRK1A in comparison to other targets. Interestingly, these results are in agreement with experimental data because boldine inhibits CK2 and DYRK1A [56,58]. Considering that scoring functions are limited, we further evaluated, in depth, the binding mode and interactions of representative compounds with the predicted target with the highest binding affinity.

Three targets (CHK1, CK2, and EHMT2) had the highest binding affinity with, at least, one compound of each group. Consequently, for these proteins, all protein–ligand complexes obtained in the docking studies were analyzed and the complex of one representative ligand of each group is discussed here. CHK1 has been experimentally validated as an anticancer target for treating OSCC in in vitro and in vivo models [55]. Coclaurine analogs presented a similar binding manner in comparison to the cocrystallized inhibitor 4,4’-(1-propyl-1H-1,2,4-triazole-3,5-diyl)bis(2,5-dihydro-1,2,5-oxadiazol-3-amine) (3C3), except for (*S*)-*N*-methylcoclaurine which was oriented outwards the ATP binding site (Figure 6). The tetrahydroisoquinolin-7-ol moiety of (*R*)-*N*-methylcoclaurine was hydrogen-bonded to E85 and C87. This group also interacted with S147 via hydrogen bond whereas the 4-hydroxyphenyl ring was positioned in a deeper region and established hydrogen bond interactions with E55, N59, and F149. Interestingly, this region was highlighted previously to be explored to develop more potent and selective CHK1 inhibitors [65].

Regarding boldine analogs, isocorydine was the only compound oriented outward to the binding site, while boldine and asimilobine were superimposed to 3C3 (Figure 6). The tetrahydro-4*H*-dibenzo[de,g]quinoline moiety of boldine was anchored at the c binding site through hydrogen bonds with E85 and E91, but this compound was also involved in other hydrogen bonds with Y20 and D148. In addition to polar interactions, both compounds were involved in van der Waals interactions with residues like L15, V23, A36, K38, and L137. Hydrogen bonds with E85 and C87 are key interactions for stabilizing inhibitors at the ATP binding site of this enzyme, and many of the interactions observed for these compounds were also described for other proven inhibitors [65]. Hence, our results indicate that these compounds could bind to this enzyme and inhibit its activity.

Also, coclaurine inhibits EHMT2 with IC_50_ of 79.43 µM [59] and, because the expression of this protein is augmented in several cancer types, we included this enzyme as a putative target of the studied fraction. Only the *R* enantiomers of coclaurine and *N*-methylcoclaurine were positioned at the lysine channel of EHMT2, like the cyclopropylethyl group of the cocrystallized inhibitor RK-701 (Figure 6). Many inhibitors of this enzyme are competitive and bind to the substrate binding site, namely, the lysine channel, and block substrate access [66]. Surprisingly, (*R*)-*N*-methylcoclaurine presented a higher Z-score with this enzyme than (*R*)-coclaurine. The tetrahydroisoquinolin-7-ol of (*R*)-*N*-methylcoclaurine was positioned at the lysine channel and was hydrogen-bonded to L1086, while the 4-hydroxyphenyl ring was involved in hydrogen bond and cation–π interactions with D1088 and R1157, respectively. In addition, boldine analogs are bound at the lysine channel. Isocorydine was hydrogen-bonded to Y1154 and R1157 and was sandwiched by T-shaped π–π stacking interactions with F1087 and F1158. Most of these interactions are also responsible for anchoring other large and small inhibitors of this enzyme [67], which suggests that these compounds could block the substrate access to the active site of EHMT2.

For CK2, boldine is a proven inhibitor of this protein with IC_50_ of 0.7 µM, and was shown to bind at the ATP binding site [10,56]. Our docking studies demonstrated that isocorydine was positioned outwards the binding site while asimilobine was bound in a deeper region, like boldine (Figure 6). This compound showed a hydrogen bond interaction with K68 and a π–sulfur interaction with M163. Likewise, *N*-methylcoclaurine enantiomers exhibited a similar binding manner while coclaurine enantiomers did not explore this region, suggesting that they do not inhibit this enzyme. (*R*)-*N*-methylcoclaurine established the same interactions with K68 and M163, in addition to the hydrogen bond with V116. Van der Waals contacts were also conserved with residues L45, V66, and I174. Since both compounds exhibited a comparable interaction profile to known inhibitors, especially the hydrogen bond with K68 [56], our results suggest that asimilobine and *N*-methylcoclaurine could block CK2 activity.

Moreover, DYRK1A is a protein already validated as an anticancer therapeutic target for treating OSCC [57] and is inhibited by boldine with an IC_50_ of 4.84 µM [58]. As boldine and its analogs exhibited positive Z-score values with this protein, we investigated their binding mode at the ATP binding site. These compounds exhibited a different binding pose when compared to the cocrystallized inhibitor 10-chloro-2-iodo-11*H*-indolo[3,2-c]quinoline-6-carboxylic acid (4E2), though they were bound at the same region (Figure 7). This is likely to occur due to the presence of a carboxylic acid group in known inhibitors that is lacking in the natural products studied here. Isocorydine was bound in a deeper region of the binding pocket, close to the conserved water molecule, which explains its higher Z-score. This compound was hydrogen-bonded to S242 which is also observed for other inhibitors. Other nonpolar contacts with residues such as I165, F170, V173, A186, V222, F238, L241, L294, and V306 were also observed. Indeed, the observed hydrogen bond, as well as many of the nonpolar contacts, have been reported as known inhibitors of DYRK1A [68] and the similar interaction network implies that boldine analogs, especially isocorydine, might also inhibit this enzyme.

Additionally, isocorydine showed one of the highest Z-score values with the LXRβ. This receptor is found in OSCC cells, and its activation led to cholesterol efflux and inhibition of cell proliferation, pointing to it as a promising therapeutic target [61]. Consequently, we investigated the binding mode of this compound and its analogs with this protein. The three analogs showed distinct binding modes, but isocorydine bound closer to H435, like the inhibitor T0901317 (Figure 7). As a result, this compound was involved in a hydrogen bond with this residue, which is critical for agonists’ binding. The tetrahydro-4*H*-dibenzo[de,g]quinoline moiety of this compound was involved in cation–π interaction with F271, in addition to other van der Waals contacts with A275, M312, L345, F349, and L442. Besides the key hydrogen bond with H435, agonists are usually anchored by nonpolar interactions, as we observed for isocorydine [69], which indicates that this compound is a potential LXRβ agonist.

Two proteins, AR and VEGFR2, showed high Z-score values with coclaurine analogs. Blockade of the VEGFR2 pathway can cause antiproliferative activity against OSCC cells, besides inhibiting angiogenesis which, in turn, decreases tumor growth [62]. Hence, docking studies with coclaurine analogs were performed within the ATP binding site of this receptor. Unlike the cocrystallized inhibitor *N*-4-(3-methyl-1*H*-indazol-6-yl)-*N*-2-(3,4,5-trimethoxyphenyl)pyrimidine-2,4-diamine (SAV), all the compounds showed an “S-shaped” conformation (Figure 7), which has been observed for other VEGFR2 inhibitors. However, the representative compound, (*R*)-coclaurine, resembled the interaction network of the proven inhibitors, especially the hydrogen bonds with Cys919, which is critical for anchoring inhibitors at this binding site [70]. In addition, (*R*)-coclaurine established other interactions, such as hydrogen bonds with E917 and V914, and van der Waals interactions with L840, V848, A866, K868, V916, F918, K919, and L1035. The similar network interaction pattern of these compounds and known inhibitors suggests that coclaurine analogs could target the ATP binding site of VEGFR2 and block its activity.

Since AR is expressed in some OSCC cells and its inhibition leads to a reduction in cell migration [54], we further analyzed whether coclaurine analogs could bind to the surface allosteric binding site activation function 2 (AF2) and block the binding of AR coactivators [71]. Both enantiomers of coclaurine and *N*-methylcoclaurine were superimposed to the cocrystallized inhibitor K10 (Figure 7). The 4-hydroxyphenyl ring of (*R*)-coclaurine was positioned at the S1 subsite, whereas the tetrahydroisoquinolin-7-ol moiety occupied S2 and S3 subsites and interacted with M734 via hydrogen bond. Other van der Waals contacts were also established. Since this compound resembled the binding mode of known inhibitors and conserved the hydrogen bond interaction with M734 [71], these derivatives are likely to inhibit AR activity and contribute to anticancer activity by inhibiting cell migration as well.

In addition to the literature reports, we used the experimental data obtained from clinical samples of patients with head and neck cancer available within the Human Protein Atlas [72] database to further validate these proteins as therapeutic targets in OSCC (Table 3). Interestingly, genes that encode most of these proteins are expressed in these samples (FPKM < 1.0). In agreement with these data, more than 25% of patients with this type of cancer exhibit medium or high expression levels of most of these proteins as well, which supports them as potential therapeutic targets in OSCC drug development.

Collectively, despite structural similarities of the compounds found in fraction 14.05, our target fishing strategy suggested that different compounds can target different proteins of OSCC cells that are clinically relevant. Therefore, our findings indicate a multitarget mechanism of action of these natural products that may also act synergistically to potentiate the anticancer activity of this fraction which, in turn, is of great interest to the development of new drugs to treat multifactorial and complex diseases such as oral cancer.

## 4. Conclusions

In summary, the antitumor activity of *Piper cernuum* dichloromethane partition (PCLd) was tested in a chemically induced oral cancer model in mice. We show that PCLd reduces tongue atypic and precancerous alterations induced by 4NQO (Figure 1), it is nontoxic to mice in tested doses (Table 1), and it significantly reduced moderate dysplasia and carcinoma (Figure 2) and extended life expectancy (*p* < 0.0001) of mice with OSCC (Figure 3). A biological activity-guided search to identify the main compounds in PCLd was made by sequential chromatographic fractioning and selection by cytotoxicity and selectivity. Fractions 09.07 and 14.05 were the most cytotoxic and selective (Table 2). Fractions 09.07 and 14.05 were not hemolytic and induced classical apoptotic phenotype (Figure 4). Substances were annotated by dereplication of mass spectrometry data on the GNPS platform, corroborating the cytotoxic activity observed in both active fractions of the *Piper cernuum* studied. The annotation of the aporphine alkaloids, isocorydine (1) boldine (2), asimilobine (3), and the benzyltetrahydroisoquinoline alkaloids coclaurine (4) and *N*-methylcoclaurine (5) stands out (Figure 5), as they are being described for the first time in the species *P. cernuum*, in addition to presenting several studies that prove its anticancer action, due to the structural characteristics of this alkaloid type, such as planarity. These compounds had their putative molecular targets determined in silico where the most relevant ones found were the androgen receptor (AR), CHK1, CK2, DYRK1A, EHMT2, LXRβ, and VEGFR2 (Figure 6). All these data are summarized in Appendix A. The present work reinforces the relevance of the phytochemical study of the species *P. cernuum*, indicating the potential development of new cytotoxic drugs of natural origin.

## Figures and Tables

**Figure 1 biomedicines-11-01914-f001:**
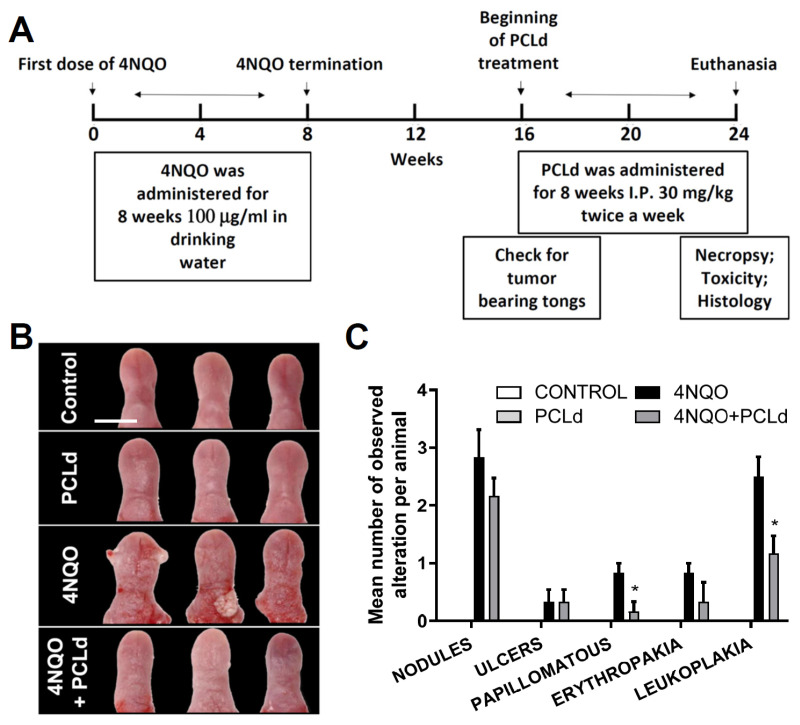
**PCLd treatment reduces macroscopic alterations in 4NQO-treated mice.** (**A**) Timeline of experiments to study chronic toxicity and macroscopic and histopathological analyses. In the first 8 weeks, 100 μg/mL of 4NQO in drinking water was administered to the 4NQO and 4NQO + PCLd subgroups. During the next 8 weeks, tumors were expected to develop, and within 8 weeks, 30 mg/kg of PCLd was administered with intraperitoneal injections in the PCLd and 4NQO + PCLd subgroups. At week 24, all subgroups were euthanized to perform macroscopic and histological analyses. (**B**) Images of 3 representative tongues of each group of mice for macroscopic analysis. Photos were taken on the day the animals were euthanized in the 24th week. (**C**) Graph of the quantification of macroscopic alterations in all animals. A chart containing the results by subgroups (control, PCLd, 4NQO, 4NQO + PCLd). A significance level of *p* < 0.05 was used to determine statistical significance following a one-way ANOVA with Tukey’s posttest analysis and was marked with a *. Scale bar: 0.5 cm.

**Figure 2 biomedicines-11-01914-f002:**
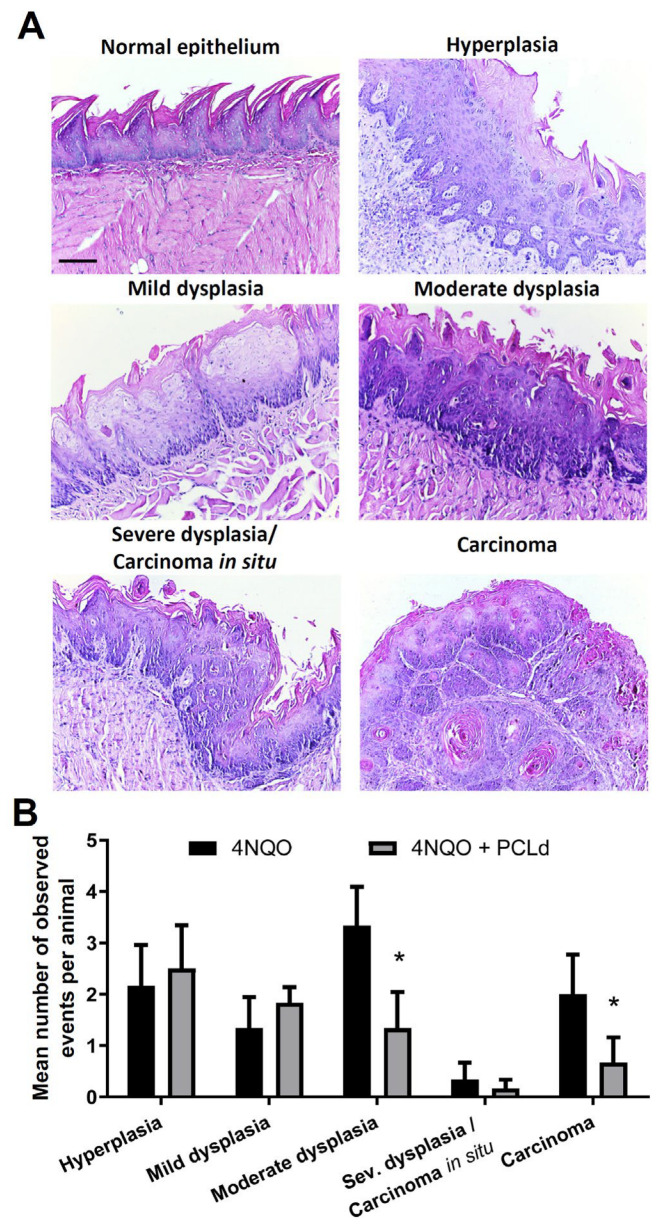
**PCLd reduces pathological characteristics induced by 4NQO.** (**A**) HE images of representative normal or atypical tissue showing normal epithelial, hyperplasia, mild dysplasia, moderate dysplasia, severe dysplasia/carcinoma in situ, and squamous cell carcinoma. Slides were analyzed, identifying the event in its greatest progression and anatomical location. (**B**) Quantification of microscopic changes: the 4NQO + PCLd group presented lesions with fewer cell atypia and severity events compared to the 4NQO group. A significance level of *p* < 0.05 was used to determine statistical significance following a one-way ANOVA with Tukey’s posttest analysis and was marked with a *. Scale bar: 100 µm. Error bars are represented.

**Figure 3 biomedicines-11-01914-f003:**
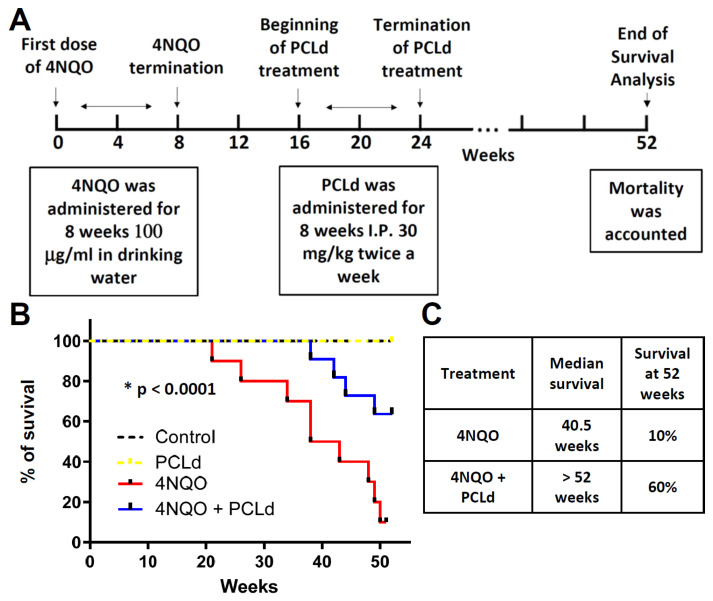
**PCLd increases the survival 4NQO-treated mice.** (**A**) Timeline of the survival experiment of mice in the control, PCLd, 4NQO, and 4NQO + PCLd groups. Each group had n = 10 mice. (**B**) Graph of the Kaplan–Meier survival curve of mice after treatment with 4NQO alone or in combination with dichloromethane fraction of *Piper cernuum* leaves, as described in (**A**). (**C**) Mice in the 4NQO group had a mean survival of 40.5 weeks, while in the 4NQO + PCLd group, the mean survival was greater than 52 weeks, indicating an increase of more than 20% in the survival of mice (*p* < 0.0001).

**Figure 4 biomedicines-11-01914-f004:**
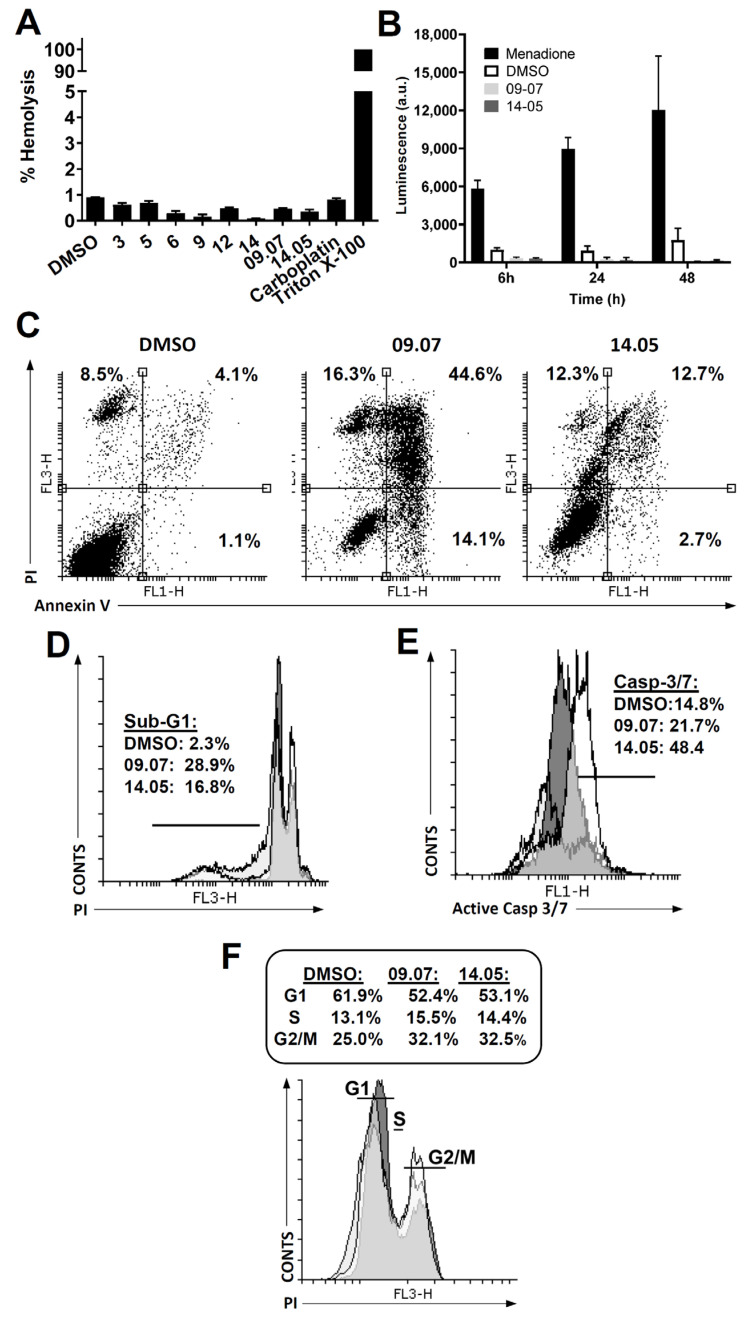
**PCLd fractions lack hemolytic activity, do not induce ROS production, and induce cell death by apoptosis.** (**A**) Hemolytic activity of PCLd fractions at 500 μg/mL. One-way ANOVA with Dunnett’s posttest was performed, in which all columns were significantly different from the control (Triton X-100) with *p* < 0.0001. (**B**–**E**) SCC9 cells were treated with 2 × IC_50_ of the PCLd fractions 09.07 or 14.05. (**B**) Luminescence assay of PCLd fractions 09.07 and 14.05 did not produce reactive oxygen species (ROS). (**C**) Representative flow cytometry data from the analysis of phosphatidylserine exposure 48 h after treatment with fractions 09.07 and 14.05. Cells were stained with FITC-conjugated Annexin-V and PI and analyzed by flow cytometry. (**D**) Representative flow cytometry of SCC-9 cells demonstrating DNA fragmentation (Sub-G1 DNA content) after treatment with the indicated fractions or DMSO for 48 h. (**E**) Representative flow cytometry data of caspase-3/7 activity. SCC9 cells were treated with the indicated fractions. (**F**) Representative flow cytometry histogram of SCC-9 cells demonstrating cell-cycle phases (G1; S; G2/M DNA-content) after treatment with the indicated fractions or DMSO for 48 h, using PI staining. Results from at least three independent experiments.

**Figure 5 biomedicines-11-01914-f005:**
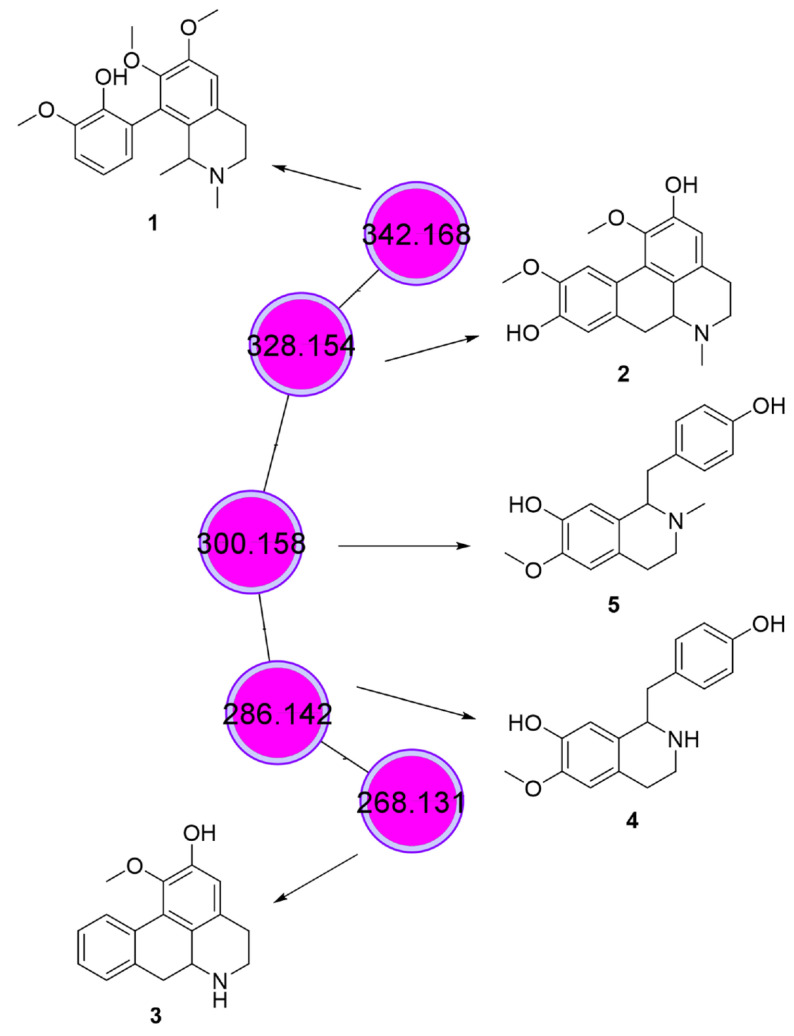
Aporphine and tetrahydrobenzylisoquinoline alkaloids annotated in fraction 14.05 by LC-MS/MS analysis, followed by GNPS. (**1**) Isocorydine, (**2**) boldine, (**3**) asimilobine, (**4**) coclaurine, and (**5**) *N*-methyl coclaurine.

**Figure 6 biomedicines-11-01914-f006:**
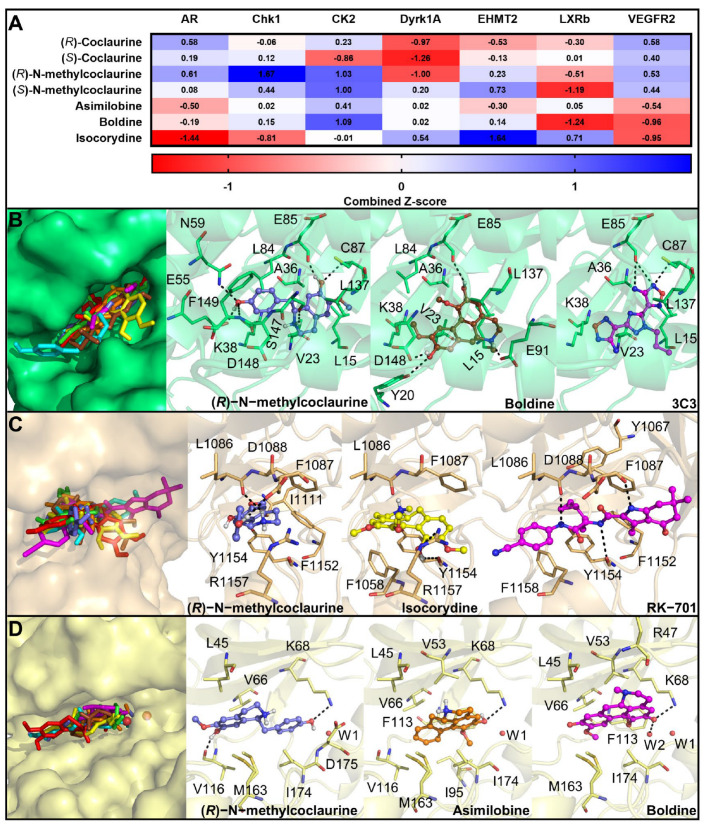
**Molecular docking results of the main alkaloids from 14.05 fraction comprising the active fraction against OSCC cells.** (**A**) Combined Z-score calculated for each compound of the 14.05 fraction with each putative target; superposition and molecular interactions of representative compounds and cocrystallized inhibitors with (**B**) CHK1, (**C**) EHMT2, and (**D**) CK2. Ligands are represented as sticks and balls and their carbon atoms are colored as follows: (*R*)-coclaurine (green), (*S*)-coclaurine (red), (*R*)-*N*-methylcoclaurine (slate), (*S*)-*N*-methylcoclaurine (cyan), asimilobine (orange), isocorydine (yellow), boldine (brown), and cocrystallized inhibitors (magenta). Hydrogen bonds are represented as dashed lines. Water molecules are shown as red spheres.

**Figure 7 biomedicines-11-01914-f007:**
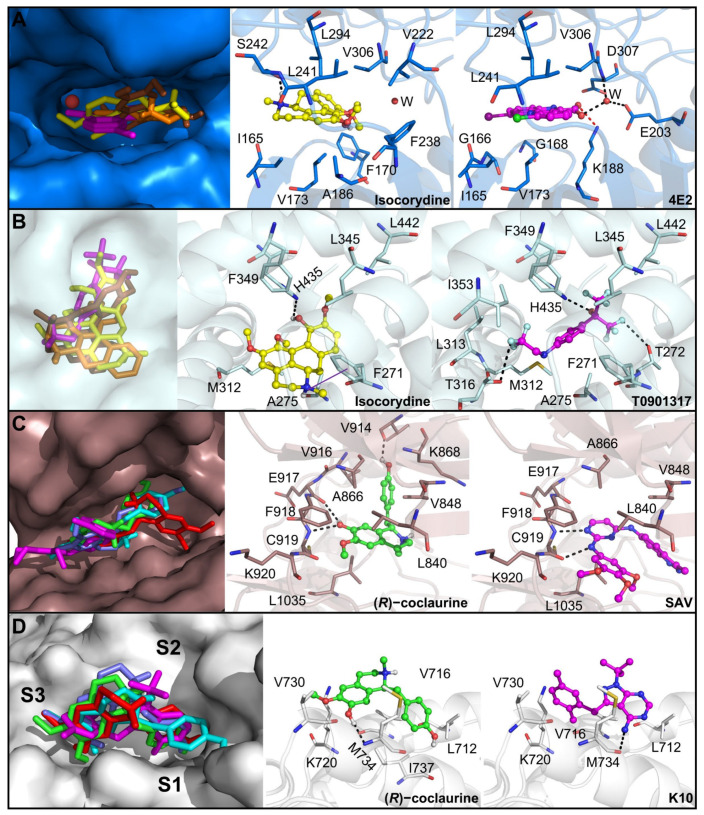
Molecular docking of the main alkaloids from active fraction 14.05 comprising the active fraction against OSCC cells with potential anticancer targets. Superposition and molecular interactions of representative compounds and cocrystallized inhibitors with (**A**) DYRK1A, (**B**) LXRβ, (**C**) VEGFR2, and (**D**) AR. Ligands are represented as sticks and balls and their carbon atoms are colored as follows: (*R*)-coclaurine (green), (*S*)-coclaurine (red), (*R*)-*N*-methylcoclaurine (slate), (*S*)-*N*-methylcoclaurine (cyan), asimilobine (orange), isocorydine (yellow), boldine (brown), and cocrystallized inhibitors (magenta). Hydrogen bonds and ionic interactions are represented as black and red dashed lines, respectively. Aromatic interactions are shown as purple lines. Water molecules are shown as red spheres.

**Table 1 biomedicines-11-01914-t001:** **Chronic toxicity study of the PCLd partition of *Piper cernuum*.** Acute toxicity study: morbidity, mortality, gross organ necropsy, and histology of mice treated as indicated. For more information and results, see Appendix A.

Treatment	Dose	Change in Body Weight	Change in Food Consumption	Morbidity ^a^	Mortality	Gross Necropsy ^b^	Histology^c^
Control	0	Absent	Absent	Absent	Absent	No alteration	Normal.
PCLd	480 mg/kg	Absent	Absent	Absent	Absent	No alteration	No significant alteration in comparison to the control.
4NQO	100 mg/mL	Absent	Absent	Absent	Absent	No alteration	Moderate/severe pulmonary arterial and venous hyperemia. Moderate portal hyperemia. Perivascular and periportal lymphocyte focus.
4NQO + PCLd	100 mg/mL + 480 mg/kg	Absent	Absent	Absent	Absent	No alteration	Moderate/severe pulmonary arterial and venous hyperemia. Moderate portal hyperemia.

^a^ Morbidity symptoms were analyzed every day, two times a day, and were considered as tremors, convulsion, salivation, diarrhea, lethargy, coma, signs of pain, and mobility defect. ^b^ Gross organ necropsy of the liver, thymus, right kidney, right testicle, heart, major lymph nodes, and lung. ^c^ Histopathology of the lung, kidney, heart, liver, and spleen was accessed by a trained pathologist and is summarized here.

**Table 2 biomedicines-11-01914-t002:** IC_50_, SD (standard deviation), and selective index (SI) of chromatographic fraction form PCLd in SCC9 and fibroblasts determined by nonlinear regression. *Piper cernuum* fractions were tested on SCC9 tumor cells and fibroblasts. ND*: value not determined, outside the tested range.

Fraction	SCC9—Oral Cancer		Primary Gingival Fibroblast		Selective Index (SI)
	IC_50_ (µM)	SD	IC_50_ (µM)	SD	
**3**	76.35	0.03	38.84	0.81	0.51
**5**	79.64	0.04	111.2	0.79	1.39
**6**	71.96	0.04	73.73	0.87	1.02
**9 ^a^**	40.25	0.06	107.50	0.03	2.67
**12**	45.08	0.04	14.00	0.03	0.31
**14 ^a^**	77.63	0.05	>600	ND	>7.7
**09.01**	71.3	0.05	103.9	0.02	1.46
**09.03**	46.26	0.04	81.94	0.02	1.77
**09.05**	47.45	0.02	74.02	0.02	1.56
**09.07**	36.87	0.01	74.71	0.02	2.03
**09.09**	52.18	0.03	74.2	0.03	1.42
**14.03**	74.8	0.02	63.77	0.02	0.85
**14.05**	64.2	0.04	162.6	0.07	2.53
**14.07**	>600	ND	239.6	0.06	ND
**14.09**	>600	ND	>600	ND	ND
**14.10**	>600	ND	>600	ND	ND
**Carboplatin**	322.30	0.04	320.50	0.07	0.99

^a^ Indicated fractions were submitted to a new round of chromatographic separation that were tested below as fractions 09.07× and 14.05.

**Table 3 biomedicines-11-01914-t003:** Putative molecular targets of annotated compounds of the active fraction 14.05 in OSCC.

Protein	PDB ID	Source	Target Association with Cancer and/or OSCC	Median FPKM ^a^	Patients with High or Medium Protein Expression Level (%) ^a^	Ref.
Androgen receptor (AR)	2PIP	PharmMapper	This receptor is expressed in different OSCC cell lines and is required for cell migration,	0.1	25	[54]
Serine/threonine-protein Checkpoint kinase 1 (CHK1)	2CGW	PharmMapper	Targeting CHK1 results in in vitro and in vivo antiproliferative activity against OSCC,	3.3	N/A	[55]
Casein kinase 2 alpha (CK2)	6HNY	ChEMBL/PBD	Boldine inhibits CK2 and induces a proapoptotic effect. Also, this protein has been validated as an anticancer target for OSCC in in vivo models,	17.5	100	[56]
Dual specificity tyrosine phosphorylation regulated kinase 1A (DYRK1A)	4YLK	PharmMapper	DYRK1A is upregulated in OSCC and is required for tumor growth and stemness,	5.6	75	[57,58]
Histone-lysine *N*-methyltransferase 2 (EHMT2)	7 × 73	ChEMBL	EHMT2 levels are increased in several cancer types. Consequently, it is considered an epigenetic target that is inhibited by coclaurine,	8.1	75	[59,60]
Oxysterols receptor Liver X receptor-β (LXRβ)	1PQ9	PharmMapper	Activation of this receptor reduces OSCC cells proliferation and tumor growth,	N/A	N/A	[61]
Vascular endothelial growth factor receptor 2 (VEGFR 2)	3CJF	PharmMapper	Inhibition of the VEGFR2 pathway induced apoptosis and suppressed angiogenesis in OSCC. In addition, a VEGFR2 inhibitor is safe and effective against OSCC in clinical trials,	2.1	25	[62]

^a^ Expression data were obtained from RNA-seq data (Median FPKM) from The Cancer Genome Atlas. Percentage of patients with head and neck cancer that present medium and high levels of protein expression were obtained from The Human Protein Atlas. N/A: not available.

## Data Availability

Data will be available upon request.

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
