# Peer review of "Anticancer Activity and Molecular Targets of Piper cernuum Substances in Oral Squamous Cell Carcinoma Models"

_biomedicines, 2023, doi:10.3390/biomedicines11071914_

Round 1

Reviewer 1 Report

The manuscript entitled “Anticancer Activity and Molecular Targets of Piper cernuum 2
Substances in Oral Squamous Cell Carcinoma Models “ presents well conducted study on potential role of PCLd as anti-cancer agent in mouse model of OSCC. The range of methods is properly chosen and the description of methods and results is adequate. The study is also well illustrated.

However, some issues have to be clarified and corrected:

First of all, the Biomedicines journal expects the authors to include the following sections of a manuscript: “Introduction, Materials and Methods, Results, Discussion”. The structure of the manuscript here is different as it does not contain separate “Discussion” section.

Minor issues:

I think that it would be beneficial for the readers to read something more about the plant from which the substance were isolated or the family Piperaceae in the introduction section.

What was the range of concentrations for IC50 calculations?

How the dose of the PCLd was established in in vivo experiments?

When the dose of 30mg/kg was used, how it is measured/weighted since the extract is suspended in methanol:water (1:1) and extracted?

In figure legends the indication of statistical significance and comparisons (“statistical difference of p < 0.05”)  should be clarified.

There are multiple sentences that sound awkward or carry grammatical errors, please rephrase/correct i.e.:

Line 204: were obtained by (ATCC) – „from”?

Line 65: showing high of cytotoxicity and selectivity against OSC

Line 159: tongue from animals were collected for macroscopic and

Line 391: is described at in Figure 3A.

Line 416: and lower complexity of according to the

Line 421: positive control Carboplatin – lower case „carboplatin”

Line 423: toxic in different human and murine tumor cells

Line 443: the controls don’t do not promote hemolysis in human red

Line 470: can produce ROS production.

Line 491: the most selective and that induce higher

There are multiple sentences that sound awkward or carry grammatical errors, please rephrase/correct i.e.:

Line 204: were obtained by (ATCC) – „from”?

Line 65: showing high of cytotoxicity and selectivity against OSC

Line 159: tongue from animals were collected for macroscopic and

Line 391: is described at in Figure 3A.

Line 416: and lower complexity of according to the

Line 421: positive control Carboplatin – lower case „carboplatin”

Line 423: toxic in different human and murine tumor cells

Line 443: the controls don’t do not promote hemolysis in human red

Line 470: can produce ROS production.

Line 491: the most selective and that induce higher

Author Response

Reviewer #1:

1) First of all, the Biomedicines journal expects the authors to include the following sections of a manuscript: “Introduction, Materials and Methods, Results, Discussion”. The structure of the manuscript here is different as it does not contain separate “Discussion” section.

Response:

We thank the reviewer for the suggestion. Although is a good suggestion we think that for this kind of paper it's more interesting to make a more concise and direct discussion together with the results (Results and Discussion). We have renamed the section “Results” as “Results and Discussion” and highlighted in the main text. While some journals have a mandatory separation between these two sections, Biomedicines (and MDPI journal overall) do not make this an obligation, as described at the Authors Instruction (https://www.mdpi.com/journal/biomedicines/instructions#preparation): “Discussion: Authors should discuss the results and how they can be interpreted in perspective of previous studies and of the working hypotheses. The findings and their implications should be discussed in the broadest context possible and limitations of the work highlighted. Future research directions may also be mentioned. This section may be combined with Results.”  

Minor issues:

2) I think that it would be beneficial for the readers to read something more about the plant from which the substance were isolated or the family Piperaceae in the introduction section.

Response:

We added the following sentence to the introduction: “Piper cernuum is a shrub native to Brazil, reaching a height of approximately 6 m and featuring large leaves measuring up to 40 cm. It is found in both primary and secondary forests and is commonly known as "pimenta de macaco" (monkey pepper). The leaves of this plant are used to prepare infusions and decoctions for the treatment of kidney and liver diseases, ulcers, colds, fevers, bronchitis, and urinary tract infections [10].”

3) What was the range of concentrations for IC50 calculations?

Response:

We now mention in the Material and Methods section the following sentence: “Each fraction was tested in at least seven different concentrations varying from 15.6 μg/mL to 1000 μg/mL.”

4) How the dose of the PCLd was established in in vivo experiments?

Response:

We have demonstrated the efficacy of PCL-d treatment in mice with chemically induced OSCC. The administered dose was both non-toxic and effective in improving the survival of the animals. Each mouse received a total dose of 480mg/kg of the dichloromethane fraction from Piper cernuum. In a previous study (Macedo et al., 2019; reference 10; https://doi.org/10.1016/j.biopha.2018.11.129), we have shown that mice can tolerate PCL-d inoculation without any detectable side effects at a single dose of 300mg/kg, and tolerate it with mild side effects at a single dose of 900mg/kg.

For our current study, we opted to administer 1/10 of the higher dose without side effects (30mg/kg per inoculation) twice a week, ensuring that the total dose did not exceed 900mg/kg. Due to the challenges in obtaining plant fractions and the large number of mice treated with this fraction (32 animals), requiring approximately 15 grams of the fraction, we were unable to achieve a higher concentration.

In summary, our findings demonstrate that treatment with PCL-d was effective in treating chemically induced OSCC in mice, with the administered dose being non-toxic and capable of increasing animal survival.

5) When the dose of 30mg/kg was used, how it is measured/weighted since the extract is suspended in methanol:water (1:1) and extracted?

Response:

Thank you for your question. We ensure that all solvents are thoroughly removed from the fractions and partitions through rotoevaporation and/or lyophilization processes. Therefore, in our study, we utilized 30mg/kg of the dry, extracted fraction of PCL-d. This approach guarantees that any remaining solvents are eliminated, and we work exclusively with the concentrated and purified form of the fraction. This is standard for any extraction process.

6) In figure legends the indication of statistical significance and comparisons (“statistical difference of p < 0.05”)  should be clarified.

Response:

Thank you for the suggestion. We updated the figure legends.

7) There are multiple sentences that sound awkward or carry grammatical errors, please rephrase/correct i.e.:

Line 204: were obtained by (ATCC) – „from”?

Line 65: showing high of cytotoxicity and selectivity against OSC

Line 159: tongue from animals were collected for macroscopic and

Line 391: is described at in Figure 3A.

Line 416: and lower complexity of according to the

Line 421: positive control Carboplatin – lower case „carboplatin”

Line 423: toxic in different human and murine tumor cells

Line 443: the controls don’t do not promote hemolysis in human red

Line 470: can produce ROS production.

Line 491: the most selective and that induce higher

Response:

Done!

Reviewer 2 Report

The author studies the effects of a chromatographic fraction from PCLd on oral squamous cell carcinoma and fibroblasts, specifically looking at its cytotoxicity and selectivity. They conduct tests on in vivo OSCC models, analyzing chronic toxicity and survival rates. Additionally, the author utilizes the GNPS platform to annotate primary compounds and perform 1H NMR analysis and explore potential molecular targets through in silico molecular docking. Albeit, I consider these findings to provide new insight into cancer-related fields, I still have some suggestions.

1, Most figures are highly professional; however, the authors should guide the readers to the meaning of the images appropriately; otherwise, it will likely cause misunderstandings. Therefore, I suggest the author consider revising these figure legends again.

2, The author characterized anticancer activity and molecular targets of Piper cernuum substances in oral squamous cell carcinoma models. However, it would be much better if the authors could provide some Workflow or Scheme for this research, I suggest that they can take a look at the recent paper in MDPI (PMID:  35563422, 36677020, 34834441)

3, In table 3, the author list putative molecular targets of annotated compounds of the active fraction 14.05 in OSCC, such as Androgen Receptor (AR), Chk1, CK2, Dyrk1, EHMT2, LXRβ and VEGFR2. However, it would be fascinating if these data could be correlated with other clinical databases. Therefore, I suggest the authors can validate their data via Proteinatlas or cBioportal (PMID: 17008526, 22588877, 32064155).

4, In Figure 1,2,4, the author may need to use other statistical analyses, such as ANOVA to calculate the P-value for three or more groups of data, and please update the “Statistical Analysis” of the Method during further revision.

5, In Figure 1-2. Please also add scale bar for image data

6, There are few typo issues for the authors to pay attention to; please also unify the writing of scientific terms. “Italic, capital”?

7, The font is too small for some of the current figures; meanwhile, the manuscript also needs English proofreading.

Editing of English language required

Author Response

Reviewer #2:

1, Most figures are highly professional; however, the authors should guide the readers to the meaning of the images appropriately; otherwise, it will likely cause misunderstandings. Therefore, I suggest the author consider revising these figure legends again.

Response:

Thank you for the suggestion. We thoroughly revised all figure legends and made several changes. However, we could not find any misleading of non-auto explicative parts. Further, no problems with figure legends were indicated by another reviewer. All information is present in the main text and figure legend otherwise in the Method section. However, if there are still any specific doubts about an experiment or result, please specify so we can readily correct them.   

2, The author characterized anticancer activity and molecular targets of Piper cernuum substances in oral squamous cell carcinoma models. However, it would be much better if the authors could provide some Workflow or Scheme for this research, I suggest that they can take a look at the recent paper in MDPI (PMID:  35563422, 36677020, 34834441)

Response:

Thank you very much for the suggestion. We made a Scheme of the research but because of the space restriction and the large amount of data already present in the work we decided to put it as Graphical Abstract (GA) and as Supplementary Figure 7 (mentioned at the conclusion).

3, In table 3, the author list putative molecular targets of annotated compounds of the active fraction 14.05 in OSCC, such as Androgen Receptor (AR), Chk1, CK2, Dyrk1, EHMT2, LXRβ and VEGFR2. However, it would be fascinating if these data could be correlated with other clinical databases. Therefore, I suggest the authors can validate their data via Proteinatlas or cBioportal (PMID: 17008526, 22588877, 32064155).

Response:

We appreciate the reviewer`s comment and included RNA-seq and protein expression levels data obtained from clinical samples of patients with head and neck cancer available within the Human Protein Atlas in Table 3. Also, we correlated these data with our predictions in section 3.5 as suggested.

4, In Figure 1,2,4, the author may need to use other statistical analyses, such as ANOVA to calculate the P-value for three or more groups of data, and please update the “Statistical Analysis” of the Method during further revision.

Response:

Thank you for the suggestion. We updated the “Statistical Analysis” of the Method describing the method used and analyzed the data with one-way-ANOVA to calculate de P-value when considered needed, as mentioned in the figure legends.

5, In Figure 1-2. Please also add scale bar for image data

Response:

Scale bar added.

6, There are few typo issues for the authors to pay attention to; please also unify the writing of scientific terms. “Italic, capital”?

Response:

We corrected all scientific terms with problems in the writing.

7, The font is too small for some of the current figures; meanwhile, the manuscript also needs English proofreading.

Response:

Figures with small fonts were revised and fonts were increased. We made a wide revision of the English.